# The Polypharmacological Effects of Cannabidiol

**DOI:** 10.3390/molecules28073271

**Published:** 2023-04-06

**Authors:** Jorge Castillo-Arellano, Ana Canseco-Alba, Stephen J. Cutler, Francisco León

**Affiliations:** 1Department of Drug Discovery and Biomedical Sciences, College of Pharmacy, University of South Carolina, Columbia, SC 29208, USA; 2Laboratory of Reticular Formation Physiology, National Institute of Neurology and Neurosurgery of Mexico (INNN), Mexico City 14269, Mexico

**Keywords:** cannabidiol, *Cannabis sativa* L., multi-target, neurological conditions, polypharmacology

## Abstract

Cannabidiol (CBD) is a major phytocannabinoid present in *Cannabis sativa* (Linneo, 1753). This naturally occurring secondary metabolite does not induce intoxication or exhibit the characteristic profile of drugs of abuse from cannabis like Δ^9^-tetrahydrocannabinol (∆^9^-THC) does. In contrast to ∆^9^-THC, our knowledge of the neuro-molecular mechanisms of CBD is limited, and its pharmacology, which appears to be complex, has not yet been fully elucidated. The study of the pharmacological effects of CBD has grown exponentially in recent years, making it necessary to generate frequently updated reports on this important metabolite. In this article, a rationalized integration of the mechanisms of action of CBD on molecular targets and pharmacological implications in animal models and human diseases, such as epilepsy, pain, neuropsychiatric disorders, Alzheimer’s disease, and inflammatory diseases, are presented. We identify around 56 different molecular targets for CBD, including enzymes and ion channels/metabotropic receptors involved in neurologic conditions. Herein, we compiled the knowledge found in the scientific literature on the multiple mechanisms of actions of CBD. The in vitro and in vivo findings are essential for fully understanding the polypharmacological nature of this natural product.

## 1. Introduction

The herbaceous plant, *Cannabis sativa* (Linneo, 1753), has been known and used for thousands of years for recreational, religious, and medicinal purposes. The most ancient record of the use of this plant as a medicine is found in the world’s oldest pharmacopeia, the Pen-ts’ao ching (China, 2737 BC) [1,2]. This plant contains many secondary metabolites including terpenoids, flavonoids, alkaloids, lignans, fatty acids, and a large number of components (approximately 120) with a hybrid biogenetic origin involving mevalonate and polyketide pathways, which are exclusive to the plant and are named phytocannabinoids (Figure 1) [3,4,5]. The two major phytocannabinoids isolated from *C. Sativa* L. are: (-)-*trans*-Δ^9^-tetrahydrocannabinol (Δ^9^-THC) and (-)-cannabidiol (CBD), (Figure 2) both of which have been studied for their therapeutic potential [6,7]. Δ^9^-THC was discovered in 1964 and is recognized as the primary psychoactive compound responsible for the behavioral effects induced by the consumption of *C. sativa* preparations, such as marijuana or hashish [8]. It has been demonstrated that Δ^9^-THC shares behavioral effects like those of other drugs of abuse in drug reinforcement animal models [9]. Moreover, *Cannabis*-related use disorder is acknowledged in the Diagnostic and Statistical Manual of Mental Disorders (DSM 5). The study of this molecule led to the further discovery of an important neuromodulatory system widely distributed throughout the brain and body, identified as the endocannabinoid system.

The endocannabinoid system is composed of (1) a network of cannabinoid receptors (CBR) expressed throughout the body identified as cannabinoid receptor type 1 and type 2 (CB_1_R and CB_2_R); (2) the endogenous endocannabinoid ligands, such as anandamide (AEA) and 2-arachidonoylglycerol (2-AG); and (3) enzymes required for the synthesis and degradation of the endogenous cannabinoids including fatty acid amide hydrolase (FAAH) and monoacylglycerol lipase (MAGL) [10]. Δ^9^-THC is a high-affinity partial agonist of both CB_1_R and CB_2_R receptors [11]. Currently, the FDA has approved three synthetic formulations of Δ^9^-THC and a Δ^9^-THC derivative with clinical application (Table 1) [12]. Side effects of these drugs are related to intoxication and include sedation, confusion, dysphoria, and paranoia, among others [13]. Unlike Δ^9^-THC, the second major phytocannabinoid, CBD, does not induce intoxication, and it does not exhibit the characteristic profile of a drug of abuse, although it has some psychoactive properties potentially relevant to its therapeutics usefulness. For this reason, CBD seems to be a more promising therapeutic compound than Δ^9^-THC. However, in contrast to this last one, our knowledge of the neuro-molecular mechanisms of CBD is limited, and its complex pharmacology has not been fully elucidated yet.

CBD is a phenolic monoterpene that was first isolated in 1940 from Mexican marijuana (*Cannabis sativa* L.) by Adams, Hunt, and Clark [14] and from the resin of *Cannabis indica* (syn. *Cannabis sativa* L.) by Jacob and Todd [15]. Later, in 1963, Mechoulam and Shvo [16] established the structure of CBD from Lebanese hashish. CBD is a hydrophobic drug, with absorption into the bloodstream occurring in the intestine. CBD is rapidly absorbed by adipose tissue and other organs and passes through the blood–brain barrier into the central nervous system (CNS). Unfortunately, the bioavailability of CBD is low when it is orally administered (6–19%); thus, the administration of 10 mg of CBD reaches the maximum concentration in plasma ~3 μg/L at 2.8 h [17]. The administration of CBD by different routes, across a wide range of doses, does not induce serious side effects or toxicity in humans or other species [18]. Repeated administration over a month to healthy volunteers (daily doses ranging from 10 to 400 mg) did not induce any significant abnormalities in neurological, psychiatric, or clinical exams [19]. Thus, it has been concluded that CBD possesses a desirable safety profile [20]. While the CBD pharmacokinetic profile has been well established, its pharmacodynamics have been more complicated to elucidate [21].

CBD exerts a broad-spectrum pharmacological effect on several conditions such as pain, inflammation, epilepsy, and anxiety, among others, supporting the therapeutic potential use of CBD in the treatment of diverse neurodegenerative diseases and neuropsychiatric conditions [22,23]. In rodent models, CBD has been shown to have anxiolytic [24,25], anti-depressive [26,27], and anti-inflammatory effects [28], among others. The study of the pharmacological effects of CBD has been growing in recent years, and, therefore, new and exciting findings are discovered gradually. The mechanisms of action for CBD to exert its effects are complex. CBD does not seem to interact with specific CBR [29], but it has been reported that it interacts with around 56 molecular targets, including ionotropic receptors, nuclear receptors, metabotropic receptors, and enzymes, among others [21]. Due to the broad spectrum of possible medical applications of CBD, many naturally occurring and synthetic CBD derivatives have been described in the scientific literature and numerous patents for its potential clinical use [30]. Recently, the FDA approved the use of pure CBD (Table 1) [12,31,32,33] for the treatment of medically refractory seizures in patients with Dravet syndrome (DS) or Lennox–Gastaut syndrome (LGS), conditions more prevalent in children. There are currently ongoing clinical trials for the use of CBD in other conditions such as autism [34,35], schizophrenia [36], and other diseases. Due to its multi-targeted properties, CBD has been shown to have enormous therapeutic potential while lacking psychotropic adverse events [37,38]. This work aims to review and integrate the different mechanisms of action of CBD on its molecular targets and its pharmacological implications in animal models and humans.

**Table 1 molecules-28-03271-t001:** FDA-approved cannabis-based pharmaceutical-grade drugs developed and marketed to 2023 [12,31,32,33].

Drug Name	Basic Formulation	Indication
Marinol^®^/Syndros^®^	Dronabinol (synthetic Δ^9^-THC)	Appetite stimulation. Antiemetic associated with chemotherapy
Cesamet^®^	Nabilone (synthetic Δ^9^-THC derivative)	Antiemetic associated with chemotherapy
Epidiolex^®^	CBD Plant-derived	Dravet syndrome and Lennox–Gastaut Seizures associated with tuberous sclerosis complex (TSC)
Sativex^®^ Not FDA-approved Registered for commercial distribution in Europe and Canada	CBD: Δ^9^-THC 1:1 Plant-derived	Pain relief Spasticity related with multiple sclerosis (MS) Tourette syndrome (frequency and severity of motor and vocal tics)

## 2. Molecular Targets Reported for Cannabidiol

CBD is a complex multi-target molecule, meaning that it can exert different pharmacological effects by interacting with highly diverse molecular targets. CBD behaves as an agonist (Table 2), inverse agonist, or antagonist (Table 3) on different receptors. CBD can also behave as an allosteric negative (NAM) or positive (PAM) modulator. CBD also exerts an effect on several enzymes, both neuro-enzymes and hepatic ones. CBD’s effects have been reported to vary across concentrations and doses in vitro and in vivo models (Figure 3). First, we review the effects of CBD on ion channels, followed by the effects of CBD on metabotropic receptors, concluding with the effects of CBD on neuro-enzymes that metabolize and regulate neurotransmission systems. Because of its relevance, the literature about the effects of CBD on hepatic enzymes is also briefly considered. This section will be divided by subheadings and provide a concise and precise description of the experimental results, their interpretation, as well as the experimental conclusions that can be drawn.

### 2.1. Ligand-Gated Ion Channels

Glycine receptors (GlyRs) are ligand-gated chloride ion channels that mediate fast inhibitory neurotransmission [53]. GlyRs are expressed in the spinal cord and the brain stem where they are mainly involved in motor control, sensorial processing, and pain perception [54]. GlyRs are pentameric membrane proteins composed of four isoforms of the α subunits (α1–4) and a single isoform of the β subunit [55]. It has been demonstrated that high-range concentrations of CBD directly activate the strychnine-sensitive GlyRs, while at low micromolar concentration range, CBD exerts a positive allosteric modulation (PAM) of GlyRs [56]. GlyRα3 is highly expressed in the superficial layer of the spinal dorsal horn and is involved in the antinociceptive process. It has been reported that CBD can suppress persistent inflammatory and neuropathic pain by targeting these receptors in animal rodent models for pain [40].

GABA-A receptors (GABA-_A_Rs) are ligand-gated ion channels that mobilize chlorine anion, producing hyperpolarization of cells and a subsequent reduction of neuronal activity through the actions of the amino acid GABA, which is the major inhibitory neurotransmitter in the mammalian brain. GABA-_A_Rs are formed by combinations of 19 subunits: 6 alpha, 3 beta, 3 gamma, 3 rho, and one each of delta, epsilon, pi, or theta, thus generating a wide variety of isoforms. GABA-_A_R exhibit multiple allosteric binding sites. Pharmacological interactions with GABA-_A_R are extremely complex [57]. It has been described that in GABA-_A_R transfected into *Xenopus laevis* oocytes, CBD is capable of increasing GABA-_A_R mediated currents in a dose-dependent manner, with the β subunit being the main binding site for it. Therefore, CBD is a PAM of GABA-_A_R in micromolar concentration ranges [58]. This mechanism of action seems to be related with the anxiolytic and anticonvulsant effects exerted by CBD [41].

The only ionotropic serotonin receptors are 5-HT_3_ ionotropic serotonin receptors (5-HT_3_Rs). They are ligand-gated cation channels located in the CNS and PNS [59,60]. The 5-HT_3_R is expressed in regions that are involved in the integration of the vomiting reflex, bradycardia, hypotension, pain transmission, analgesia, mood disorders, and control of anxiety. Meanwhile, peripheral receptors participate in the regulation of sensory transmission and autonomic functions [61,62,63]. CBD has been reported to act as a negative allosteric modulator (NAM) for 5-HT_3_R and a non-competitive inhibitor of the function of human and mice 5-HT_3_R expressed in HEK293 cells [63] and *Xenopus laevis* oocytes [64].

Five classes of nicotinic receptors nAchR subunits (α, β, y, ε, δ,) lead to different functional isoforms of the homopentameric receptor or heteropentameric receptors [65]. Homomeric α7 nicotinic acetylcholine receptors (α7-nAChR) are pentameric calcium (Ca^2+^) channels highly expressed in the nervous system and spinal cord. In the brain, α7-nAChR are distributed post- and presynaptic throughout the cortex, thalamus, and hippocampus. Both excitatory and inhibitory synaptic transmission can be modulated by these receptors [66]. In humans, a reduction in α7-nAChR expression has been associated with an increase in seizure susceptibility [67]. α7-nAChR agonists have pro-cognitive effects, and its modulation has been proposed to be relevant for the treatment of Alzheimer’s disease (AD) or cognitive symptoms of schizophrenia, among others [68]. CBD can inhibit α7nAChR in a dose-dependent manner and induce reduction of acetylcholine-evoked currents amplitude in in vitro BOSC-23 cells and in ex vivo patch clamp assays using rat hippocampal slices, while Δ^9^-THC does not affect this channel. Therefore, CBD acts as a NAM in the closed and desensitized states of these channels [69,70].

Voltage-gated sodium channel (VGSC or Nav) genes are relevant to epilepsy in humans. These Nav are transmembrane ion channels that allow the passage of sodium ions (Na^+^) along their electrochemical gradient [71]. After the activation of these channels, Na^+^ flow into the intracellular environment, with an opening of the channel for a few milliseconds, subsequently leading to its inactivation and closing of the channel, preventing a further flow of sodium ions [71]. The passage of Na^+^ through Nav channels generates transient sodium currents that produce action potentials in cardiac muscle, skeletal muscle, and neurons [72]. Mutations in the VGSC SCN1 gene led to the loss of function of the Nav_1.1_ channel, causing the development of DS. In a mouse model of Scn8a-associated epilepsy (encoding Nav_1.6_), CBD has shown efficacyin reducing seizures frequency at a dose of 320–360 mg/kg [73]. Similarly, it is suggested that refractory epilepsies may be associated with mutations in the SCN2A gene [74]. Nav channels have been the most studied targets of CBD since, in clinical studies, CBD (20 mg/kg/day) efficacy has been shown against drug-resistant seizures in DS [72]. The inhibitory mechanism of CBD is associated with the inhibition of the Nav channels on both rest and inactivate states, which suggests that CBD does not inhibit direct interaction with a specific binding site [72]. Similarly, it has been shown in vitro that CBD had no specificity for any of Nav channels (1.1–1.7), mNav_1.6_, or bacterial homomeric Nav channel (NaChBac) and voltage-gated potassium channel subunit Kv_2.1_ [71,72].

T-type voltage-gated calcium channels (VGCCs) are a family of channels highly expressed in neurons where they modulate neuronal excitability and, in other tissues, low-threshold calcium spiking or cardiac pacemaker activity. These channels have been associated with the regulation of epilepsy, sleep, and pain; however, the mechanism underlaying these effects are unclear. The endogenous cannabinoid AEA has been suggested as an endogenous inhibitor of these channels through a non-CB receptor-mediated mechanism [75]. In an in vitro model with HEK 293 cells expressing VGCC channels, it has been proposed that CBD is an inhibitor of the cation currents recorded by patch clamp for Ca_V3.1_, Ca_V3.2_, and Ca_V3.3_ channels [75].

Voltage-dependent anion-selective channel protein 1 (VDAC1) is a mitochondrial channel expressed in the outer membrane, where it has an important role in the control of cellular energy and metabolism, by acting as a regulator of metabolite transfer between the cell cytosol and mitochondria. CBD has been proposed as a direct inhibitor of the conductance of this channel by altering cytosolic calcium homeostasis, mitochondrial morphology, and function, as well as viability, which is associated with the strong immunosuppressive response and its anticancer effects [76].

### 2.2. Transient Receptor Potential Channels (TRP)

TRP channels are present in mammals and expressed in multiple body tissues, being of great importance in peripheral neurons to transmit nerve impulses produced by chemical and physical stimuli to the brain [77]. Such stimuli can be mechanical stress, heat, variation in pH, osmotic pressure, and compounds derived from plants that lead to the activation of these receptors and the consequent mobilization of cations such as K^+^, Na^+^, Mg^2+^, and Ca^2+^ [78,79]. TRP channels are classified into six families with 27 different channels, being canonical (TRPC), ankyrin (TRPA), polycystin (TRPP), mucolipin (TRPML), melastatin (TRPM), and vanilloid (TRPV) [46]. Six of these channels, TRPV1, TRPV2, TRPV3, TRPV4, TRPA1, and TRPM8, are of importance because they are activated or inhibited by cannabinoids and have been called ionotropic CBR [79]. These six channels are involved in important physiological processes such as immune function, regulation of neurotransmitter release, temperature sensation, and pain [80].

Transient receptors potential vanilloid (TRPV), and especially type 1 (TRPV1), is the most studied of the TRP channels, which was discovered to be activated by vanilloid agonists such as capsaicin, which subsequently leads to channel desensitization and a quiescent analgesic effect [46,81], which earned its discoverer the 2021 Nobel Prize in Medicine. CBD is a full, but not potent, agonist of this type of channel [46,77,81]. This mechanism of action is related to the anxiolytic, anti-hyperalgesic, and anti-inflammatory effects of CBD in animal models [62]. In addition, CBD activates the phagocytic capacity of microglia concentration-dependent (0.1–10 µM) by mobilizing Ca^2+^ dependent on TRPV1 and TRPV2. The latter has been proposed as an advantage in favoring the clearance of β-amyloid and reducing problems in patients with AD [45]. TRPV2 shares a sequence identity of 50% and similar desensitization with TRPV1. However, this receptor is insensitive to capsaicin and acts as a heat sensor: activating above 52 °C, it is associated with chronic pain and inflammation [46]. It has recently been suggested that CBD activates TRPV2 by binding in a small hydrophobic gap between the S5 and S6 helices of adjacent subunits, which have not been identified in other TRP channels [82,83]. TRPV3 is also a warm temperature sensor that activates in the range of 33–39 °C. This channel is widely expressed in the brain, skin, and tongue. It has been described that CBD produces a similar response in TRPV3 to that of its agonist carvacrol since it activates the channel but subsequently desensitizes it [44]. However, the response of CBD to TRPV3 is lower than for TRPV1 and TRPV2. It has been suggested that differences in sequence homology at the putative CBD binding site could be responsible for the low response [46]. TRPV4 is also a warm temperature sensor in the range of 25–34 °C. This channel is present in the skin where it has an important role in barrier functions and nociception. CBD has a poor response compared to the three previously mentioned TRPV channels [44,46,81].

Transient receptors potential ankyrin type 1 (TRPA1) is a sensor of low temperatures (<17 °C). It is present in the PNS, where it plays a role as a response sensor to cold, as well as hypersensitivity to cold and hyperalgesia to cold, so this receptor is important for detection of inflammatory and pain stimuli [80]. The canonical agonists of TRPA1 are isothiocyanates that are present in onions, mustard, and garlic [80]. CBD has been shown a more potent agonist than allyl isothiocyanate [44,46,81]. It was previously demonstrated in dissociated vagal afferent neurons that CBD generated its response to increasing intracellular calcium through TRPA1 [84].

Transient receptors potential melastatin 8 (TPRM8) is a temperature sensor that is activated at ~27 °C and is expressed in sensory neuron subpopulations of the PNS, where it has been found to act in the development of neuropathic pain and migraines [85]. TRPM8 agonists include icilin, eucalyptol, and menthol, which have been suggested to activate the channel at different sites [46,80]. However, CBD is an effective TPRM8 antagonist regardless of the type of agonist used [46,80].

**Table 3 molecules-28-03271-t003:** CBD such as an antagonist, NAM, or inverse agonist of receptors in the nervous system.

Receptor Type	Activity	IC_50_ (µM)	Disease Model	Tissue Expression	Cited
5-HT_3A_	NAM	0.6	LiCl-induced nausea in rats.	CNS and PNS.	[62,86]
α7nAChR	Antagonist	11.3	Inflammation in mice.	PNS, CNS (cortical, thalamic, and hippocampal regions), and skeletal neuromuscular junction.	[70,87]
Na_V1.1–1.7_	Antagonist	1.9–3.8	Drug-resistant seizures in DS models.	CNS and peripheral neurons.	[43,72]
Kv2.1	Antagonist	3.0	Epilepsy in humans and microcephaly induced in zebrafish.	Hippocampal and cortical pyramidal neurons.	[72,88,89]
GPR3	Inverse agonist	1	-	-	[90]
GPR6	Inverse agonist	0.1	-	-	[90]
GPR12	Inverse agonist	10	-	-	[90]
Ca_V3.1_	Antagonist	0.82	-	Widespread expression in neuronal and other tissue.	[75]
Ca_V3.2_	Antagonist	0.78	-	Widespread expression in neuronal and other tissue.	[75]
Ca_V3.3_	Antagonist	3.7	-	Widespread expression in neuronal and other tissue.	[75]
TRPM8	Antagonist	0.06	Rat behavioral model of headache and hind paw and cutaneous facial allodynia.	Sensory neuron subpopulations of the PNS, and circuits related to migraine pathogenesis.	[85,91]
CB_1_R	NAM	0.2 **	Seizures in cobalt-epileptic rats.	Amygdala, olfactory bulb, cerebellum, hippocampus, basal ganglia, and neocortex.	[43,92]
CB_2_R	NAM	0.24 **	-	Cells of the immune and hematopoietic system.	[43,92]
µ-OPR	Antagonist	8–12	Drug abuse, mood disorders, and pain perception models.	Amygdala, spinal cord, substantia nigra, hypothalamic nuclei, hippocampus, and dorsal root ganglia.	[43]
δ-OPR	NAM	-	Drug abuse, mood disorders, and pain perception models.	Amygdala, spinal cord, substantia nigra, hypothalamic nuclei, hippocampus, and dorsal root ganglia.	[43]
GPR55	Antagonist	0.44	Epilepsy in mouse model of DS.	Excitatory neurons of dentate gyrus in hippocampus.	[43]

CNS = Central nervous system; PNS = Peripheral nervous system; DS = Dravet syndrome; LiCl = Lithium chloride; NAM = Negative allosteric modulator. ** Ki for human CB_1_ and CB_2_.

### 2.3. Metabotropic Receptors

Cannabinoid receptors (CB_1_R and CB_2_R) share 44% of their molecular structure, and both are coupled to Gi/o protein, which negatively modulates adenylyl cyclase. However, they differ in their specificity, function, and their pattern of distribution, as well as in cellular expression. CB_1_R is highly expressed in the brain; meanwhile, CB_2_R is found predominantly in immune cells, although recent evidence demonstrates that CB_2_R is expressed in microglia and also in neurons in the brain [93,94,95,96,97,98]. These receptors mediate the physiological actions of the endocannabinoids and the behavioral effects of the phytocannabinoid Δ^9^-THC [99]. It has been reported that CBD has a weak binding affinity for CBR (Ki ≥ 10 µM) [100], but it seems that CBD is capable of modulating some of its actions. It has been reported that CBD behaves as a non-competitive NAM of CB_1_R and CB_2_R, reducing the efficacy and potency of Δ^9^–THC and other cannabinoid receptors agonists, such as synthetic cannabinoids, CP–55,940 and WIN55,212 and of the endocannabinoids, AEA and 2-AG [101,102,103,104,105].

Serotonin receptors are classified into seven families (5-HT_1-7_R) with at least 14 distinct receptor subtypes. Except for the ligand-gated ion channel 5-HT_3_R, all serotonin receptors are classical seven-transmembrane GPCRs that mediate their effects on different secondary messenger enzymes via activation of distinct G-proteins [106]. Serotonin receptors are widely expressed in multiple brain regions, and specific neurons can express several different serotonin receptors. They play a role in many physiological processes, including thermoregulation, respiration, circadian rhythm (sleep-wake cycle), vascular function, emesis, cognition, and regulation of emotion [107]. Evidence has shown that CBD interacts with 5-HTRs, in particular, with 5-HT_1A_R and 5-HT_2A_R [108]. The 5-HT_1A_R is located presynaptically and postsynaptically and therefore can act as autoreceptor and heteroreceptor, where they exert their effects through G_αi/o_ proteins to inhibit adenylyl cyclase. Studies have shown that CBD acts as an agonist with modest affinity at the human 5-HT_1A_R [108]. One of the most conclusive pieces of evidence of the pharmacological effects of CBD is its anxiolytic property, because CBD produces the anxiolytic response through a wide range of concentrations. One of the proposed mechanisms is the interaction with the 5HT_1A_R [24,108,109,110,111]. Anxiolytic effects of CBD are induced with a bell-shaped dose–response curve when administered directly into the dorsolateral periaqueductal gray, an effect mediated by the 5HT_1A_R [24]. On the dorsal raphe nucleus, CBD acts as an indirect agonist of the somatodendritic 5-HT_1A_ autoreceptors, contributing to the anti-emetic effect [112]. The 5-HT_2A_R is expressed mainly in the cerebral cortex, olfactory bulb, and brainstem nuclei. These receptors are coupled via Gq and are known as presynaptic and postsynaptic on serotoninergic terminals. CBD acts as a partial antagonist of 5-HT_2A_ [108].

Adenosine produces its physiological response by activating four G-protein coupled receptors (A_1_, A_2A_, A_2B_, and A_3_ receptors), and they are found widely distributed in most of the body tissues, participating in a large variety of pathophysiological responses, such as vasodilation, pain, and inflammation [113]. It has been demonstrated that CBD activates the A_1A_R; this mechanism is related to the capacity of CBD to suppress ischemia-induced ventricular arrhythmias [49].

Opioidergic compounds interact with opioid receptors (μ, δ, and κ receptors) and play an important role in diverse physiological and pathophysiological processes, including analgesia, respiratory depression, and psychiatric illness. They are also expressed in the cardiovascular and immune system [114]. CBD behaves as PAM at and δOPR, and it is capable of accelerating µOPR mu agonist dissociation from the binding site, thus reducing its activity (demonstrated by kinetic binding studies) [115,116].

GPR55 is a receptor that is commonly expressed in association with CBR in the brain, PNS, and other tissues such the immune system cells and microglia [117]. This receptor has been associated with different diseases such as vascular functions, motor coordination, metabolic disorders, bone physiology, pain, and cancer. Its endogenous ligand is lysophosphatidylinositol (LPI). Once activated, GPR55 can interact downstream with Gαq/11, Gα12, Gα13, or Gα12/13, depending on the tissue or cell type [118]. CBD is an antagonist of GPR55 since it can block the effect of CP55940 in cells transfected with GPR55 in vitro [118]; CBD also acts in other tissues [119,120]. GPR18 has shown low sequence homology concerning CB_1_R, CB_2_R, and GPR55 receptors; its endogenous agonist is *N*-arachidonoyl glycine (NAGly). GPR18 has been described in various tissues such as lymphoid tissue, brain, lungs, ovary, and testis, where it has been associated with sperm physiology, metabolism, and with diseases such as cancer, intraocular pressure, and pain [121]. CBD acts as an antagonist of GPR18, shown to inhibit the migration of BV-2 microglia and transfected HEK293-GPR18 cells induced by NAGly and ∆^9^-THC [122]. GPR3, GPR6, and GPR12 are three receptors with about 60% molecular sequence similarity to the CB_1_ and CB_2_ receptors. These three receptors are expressed in the reproductive system and the brain and constitutively activate adenylate cyclase through Gαs proteins. These receptors are involved in the formation of synaptic contacts as well as in differentiation and neuronal growth [123]. The GPR3 is expressed in the nervous system: in dorsal root ganglia neurons, hippocampus, amygdala, cortex, and habenula [123]. Additionally, GPR3 is expressed in the ovary, testis, skin, adipose tissue, heart, liver, breast, and eye. GPR3 has been shown to prevent apoptosis in neurons and is associated with the development of neuropathic pain, emotional disorders, and morphine-induced antinociception [124]. Similarly, GPR12 is expressed in the limbic system and associated with emotion, behavior, and memory [123]. CBD has been shown to act as an inverse agonist for all three receptors: GPR3, GPR6, and GPR12 [125]. Although, CBD showed a weak to moderate response to GPR3 [124].

### 2.4. Nuclear Receptor: Peroxisome Proliferator-Activated Receptor γ (PPARγ)

PPARγ has been identified in adipose tissue and macrophages and has been involved in glucose energy metabolism and lipid storage [52]. In general, PPARγ ligands have shown anti-inflammatory activity, and CBD acts as an agonist of this receptor [50,52,126,127]. CBD has been shown to activate PPARγ in multiple sclerosis (MS) models [51]; also, CBD prevents neurodegeneration in a rat model of AD by reducing pro-inflammatory molecules and stimulating hippocampal neurogenesis [50]. CBD also reduces VCAM-1 level and the permeability produced by ischemia in a model of the blood–brain barrier [127].

### 2.5. Enzymes

Between the neuro-enzymes that have been shown interaction with CBD, we can highlight (Table 4) acetylcholinesterase (AChE), butyrylcholinesterase (BChE), fatty acid amide hydrolase (FAAH), and arylalkylamine *N*-acetyltransferase (AANAT). Current medications to treat patients with AD are based on blocking cholinesterase enzymes. However, these drugs have been shown to have side effects such as vomiting and nausea, as well as limitations by not being able to control neuroinflammation, oxidative stress, and amyloidogenesis [128]. CBD and Δ^9^-THC can inhibit AChE, while CBD only inhibits BChE.

FAAH is a membrane protein belonging to the serine hydrolases family; this enzyme is part of the endocannabinoid system, and its main role is terminating the signaling of bioactive lipids known as fatty acid amides (FAAs) present in the CNS and peripheral tissues; this includes hydrolysis of the AEA [129]. Inhibition of FAAH activity leads to an increase in the concentration of AEA, which, when interacting with its receptors, increases neuronal transmission to reduce pain, neuroinflammation, anxiety, and depression and counteracts nicotine addiction [129]. CBD is reported to inhibit the activity of the FAAH enzyme [91,100], although this inhibition is moderate [130].

Hepatic enzymes that include cytochromes P450 (CYP) are part of a large family of hemeprotein liver enzymes classified into families or subfamilies depending on their amino acid sequence homology and are the enzymes responsible for the first step in the metabolism and biotransformation of endogenous substrates, chemicals, and drugs [131]. It has been described that CBD inhibits cytochrome P450-mediated drug metabolism; for instance, CBD increases the plasma half-life of drugs such as hexobarbital, which is metabolized by CYP2C9 in patients. CBD has recently been reported to inhibit the catalytic activity of the liver enzymes CYP1A1-2, CYP1B1, CYP1B6, CYP2C9, CYP2C19, CYP2D6, CYP2E1, CYP3A4-5, CYP3A7, UGT1A9, and UGT2B7 in in vitro models [132,133,134,135,136,137,138,139].

**Table 4 molecules-28-03271-t004:** Neuro and hepatic enzymes inhibited for CBD.

Enzyme Type	Activity	IC_50_ (µM)	Cited
AChE	Antagonist	48.1	[128]
BChE	Antagonist	36.8	[128]
FAAH	Antagonist	15.2–27.5	[91,100]
AANAT	Antagonist	<1.0	[140]
CYP1A2	Antagonist	<1.0	[132]
CYP2B6	Antagonist	1.0	[132]
CYP2E1	Antagonist	1.0	[132]
CYP3A4	Antagonist	<1.0	[132]

## 3. Therapeutic Evidence Involving CBD

Dose-dependent biphasic effects of several cannabinoids (phytocannabinoids, synthetic and endogenous) have been reported in several responses, such as motor activity [141,142,143], feeding behavior [144,145], sexual behavior [146,147], and anxiety responses [148,149], among others. Similarly, CBD has a biphasic profile on anxiety [150,151]. This should be considered in order to understand the implication of this drug in therapeutics. CBD emerges as a promising therapeutic drug associated with its multi-target activity, which, like other natural products, presents a complex polypharmacology. The high acceptance by people of the use of natural products or their derivatives may be a possible factor to obtain better results in treatment with cannabinoids associated with inflammatory diseases and for chronic diseases requiring very long-term treatments [152]. CBD is a promising molecule in therapeutics of different conditions. Because its consumption is safe and does not induce intoxication, the industry sells CBD mostly as an oil. However, CBD modifies the metabolism of other pharmacological treatments. Considering that CBD is used more as an adjuvant in the treatment of chronic conditions, such as MS or epilepsy, warning of possible pharmacological interactions among treatments should be considered. Regarding the therapeutic role that CBD has demonstrated, we will try to integrate into this discussion the results in vivo, in vitro, and in clinical trials, and then we will discuss the relationship between the activation and inhibition of the multiple receptors identified as targets of CBD (Table 5).

### 3.1. CBD Multi-Targets Promoting Anticonvulsant and Antiepileptic Effects

The modulation of ionic currents of ion channels such as GABA_A_R [41], Nav [71,72], and less documented Cav [75] might be important in the anticonvulsant and antiepileptic effect of CBD. First, the combined effect of CBD with clobazam (CBD acts as PAM on GABA_A_R) [41] has an important role in the suppression of common epileptic events by maintaining homeostasis over brain excitation and its importance for treating drug-resistant epilepsy caused by mutations that alter the channel [170]. Secondly, the target of CBD to treat DS is the non-specific inhibitory effect on the mutated sodium channels Nav_1.1_ and Nav_1.2_ [73,74]; although, it is also capable of inhibiting other sodium channels such as Nav1.3, Nav1.4, Nav1.5, Nav1.6, and Nav1.7 [71,72]. However, a very important factor that enhances the inhibition of GABA_A_ and sodium channels by CBD is its inhibitory effect on CYP1, CYP2, and CYP3 liver enzymes, such as CYP3A4 and CYP2C19 enzymes, which are necessary for the degradation of anticonvulsant drugs such as clobazam and its metabolite *N*-desmethylclobazam, leading to an increase in their T_max_ values and plasma half-life [41]. In addition, the antiepileptic effect of CBD is associated with the inhibition of the GPR55 receptor, thus increasing the release of neurotransmitters as shown in an epilepsy mouse model [169] and by reducing the expression of the CB_1_ receptor [163], although their mechanism of action is not clearly defined for these two receptors (Figure 4).

### 3.2. CBD Multi-Targets Implicated in Inflammatory and Immunosuppressive Process

Anti-inflammatory and immunosuppressive effects of cannabis have been known since 3000 years ago, when it was used as an antipyretic and to treat rheumatism [190]. Recently, in vitro and in vivo models have suggested that CBD seems to produce these effects by acting directly on the cells and tissues of the immune system [32,152], where CBD acts as an agonist of TRPV1 in vitro activating phagocytosis in microglia [45] and in vivo decreasing inflammation in both inflammatory bowel disease and allergic contact dermatitis models [175,176]. CBD is an agonist of A_2A_R, showing reduction of inflammation in a murine model of acute lung injury and inflammatory response induced for lipopolysaccharide (LPS) [174]. Additionally, in human sebocytes, CBD induces an anti-inflammatory pathway dependent on A_2A_R cAMP-TRIB3 and the subsequent inhibition of NFkB [180].

CBD also induces activation on the PPARγ acting in the inhibition of keratinocyte proliferation avoiding inflammatory response in psoriasis [177]. However, at the same time CBD can act as an antagonist of CB_2_R [176,177], it can also attenuate M2 polarization (anti-inflammatory response) of microglia via CB_2_R, as revealed in a CB_2_R knockout mice model [179]. CBD is also known for its low inhibition to δ-OPR and µ-OPR receptors, both receptors widely known for their immunosuppressive role [178]; CBD can also modulate the activity of the α7nAChR, which is known to have rapid activation to immunosuppress synthesis and secretion of TLR4-induced TNF in mast cells [87]. Additionally, CBD antagonizes GPR18, which is activated by the association of *N*-arachidonoyl glycine (NAGly) and Δ^9^-THC on a BV-2 microglia model; thus, CBD reduces the microglial migration and cytokine expression [122,191]; CBD also inhibits the mitochondrial channel VDAC1, which has been suggested to be the therapeutic target responsible for the antiproliferative and immunosuppressive properties of CBD [76]. More experimental information is still required to specify how CBD acts through the aforementioned receptors, as well as other receptors, such as the A_1A_R, TRPV2, and TRPA, to better understand its anti-inflammatory effect.

### 3.3. CBD Multi-Targets Engage in Antinociceptive and Analgesic Properties

Preclinical research has shown that depending on the dose and route of administration, CBD alone or in combination with other drugs has an antinociceptive effect in pain models [161,162]. In sub-chronic pain models in vivo, low doses of CBD (3 mg/Kg) produced an anxiolytic and analgesic effects [163]. In a rodent model, administration of CBD into periaqueductal gray (PAG) showed a dose-dependent analgesic effect in a tail-flick test, and this effect was blocked by the use of A_1A_R, CB_1_R, and TRPA1 antagonists [166], concluding that CBD might produce antinociceptive effects at the supraspinal level, most likely interacting to several targets involved in the control of pain, including TRPA1 channels and inactivation on the endocannabinoid system [91]. Another study found that CBD reduced hyperalgesia to thermal and mechanical stimuli in two different models of persistent pain. Anti-hyperalgesic effect of CBD (20 mg/kg) was prevented by the vanilloid antagonist capsazepine (10 mg/kg, i.p.), suggesting that TRPV1 are involved in its effect either through in vitro [100] or in vivo assays [165]. Moreover, acute or chronic treatment with CBD decreases the hyperalgesia and allodynia in Parkinson’s model [166]. CBD, together with a TRPV1 antagonist, reduces L-DOPA-induced dyskinesia in a model of Parkinson’s disease by acting on CB_1_R and PPARγ and reducing the expression of the inflammatory marker’s cyclooxygenase-2 (COX-2) and NFκB [126]. Systemic and intrathecal administration of CBD can reduce neuropathic pain; this effect was significantly reduced in mice lacking the α3 subunits of GlyR. This result suggests that GlyR mediates the CBD analgesic effect in neuropathic pain models [40]. The antiallodynic effects of CBD are mediated predominantly through TRPV1 since capsazepine fully prevented CBD analgesia [167]. In studies using Paclitaxel chemotherapeutic-induced neuropathic pain, CBD decreased mechanical and thermal allodynia in mice, an effect reversed by the 5HT_1A_R antagonist [164]. The 5HT_1A_R also mediates the CBD anti-allodynic effect in diabetic (streptozotocin-induced) neuropathic pain, since the specific receptor antagonist prevented the observed effect [168]. It is remarkable that in the latest two studies here reported [164,168], the effects were not prevented with CBR antagonists. This is interesting since it suggests that the effects of CBD are also dependent on the emotionally expression state of the animal since the models change the basal physiological levels.

### 3.4. CBD Multi-Targets with Antidepressant and Anxiolytic Consequences

Regarding neuropsychiatric disorders, some of the effects are well described, but the characterization of how CBD can exert them throughout the molecular target or targets is still a matter of debate. Experimental demonstrations are required as well as the exploration of other potential targets. The anxiolytic effects of CBD have been extensively demonstrated in clinical and preclinical studies (for reviews see [192,193,194,195]). The mechanism of action throughout CBD that can reduce anxiety seems to be complex and dose dependent. Accumulated evidence indicates that acute administration of CBD produces a typical ‘bell-shaped’ dose–response curve, being anxiolytic just at low-moderate doses but not at high doses [160]. The anxiolytic effects seem to be mediated by the ability of CBD to act as a 5-HT_1A_R agonist, since the reduction in anxiety-like parameters induced by CBD is blocked by a specific 5-HT_1A_R antagonist [110,111,154,159]. It must be taken into consideration that while in vitro studies suggest CBD acts as a direct 5-HT_1A_ receptor agonist [108], in vivo studies are more consistent with CBD acting as a PAM, hence facilitating 5-HT_1A_R signaling [112]. On the other hand, it has been proposed that the lack of anxiolytic effect of the high doses of CBD is related to the activation of TRPV1 receptors, since blocking TRPV1 centrally allows for high doses of CBD to be effective [160]. Another proposed mechanism is by the indirect potentiation of AEA transmission since CBD can inhibit the FAAH enzyme [155] (Figure 5). This could be related to a recent study that demonstrated that CB_1_R (but not CB_2_R or GPR55-KO mice models) is relevant in modulating the anxiolytic actions of CBD using knockout mice for these receptors [158]. It is possible that these mechanisms and other ones can be acting in conjunction, since CBD can interact with other receptors involved in the regulation of anxiety such as GABA_A_R.

CBD has a dose-dependent antidepressant-like profile in different preclinical models (for review [196]). Although CBD can modulate several molecular targets involved in the neurobiology of depression, only two of these mechanisms have so far been explored in vivo. The involvement of 5-HT_1A_R has been consistently found [153,156,157] since the effective antidepressant dose was blocked by the pre-treatment of selective 5-HT_1A_R antagonists. Supporting this idea, the depletion of serotonin blocked the antidepressant effect of CBD [197]. The other mechanism explored is the involvement of CBR. As it was discussed previously, CBD has a low affinity for CB_1_R, but it is capable of increasing AEA levels by inhibiting its reuptake [100]. Since pretreatment with a CB_1_R antagonist also blocked some anti-depressant CBD effects [156], it is proposed that the increases in AEA levels, which in turn activate the CB_1_R, explain the results. Further supporting this proposal, pharmacological increases in AEA promote antidepressant-like effects in preclinical studies [198]. Evidence indicates that, depending on the behavioral test and treatment duration, 5-HT_1A_R or CB_1_R signaling would prevail to mediate CBD anti-depressant effects [196].

### 3.5. CBD Multi-Targets in Antipsycothic Disorders

The antipsychotic capability of CBD has been demonstrated in several clinical studies, as well as in preclinical models (for review see: [174,199,200]). However, the specific pharmacological mechanisms underlying the antipsychotic action of CBD are not entirely understood. Two opposite lines of thought are possible. On one side, despite its low affinity for CB_1_R, CBD is capable of antagonizing CB_1_R agonists at reasonably low concentrations [105]. Other studies suggest that CBD acts as a NAM of CB_1_R [101]. In any case, these actions will result in a reduction of the endocannabinoid-mediated transmission. Conversely, CBD induces an enhancement of the endocannabinoid tonus by inhibiting AEA uptake and metabolism of the FAAH enzyme [100,173,174]. In support of the antipsychotic effect of CBD mediated by antagonism of the CB_1_R, the most important argument is that the acute administration of THC (an exogenous CB_1_ agonist) can transiently exacerbate psychosis in patients with schizophrenia [201] and induce transient psychotic-like symptoms in healthy volunteers [202,203]. This latest effect can be blocked by pre-treatment with CBD [204]. Arguing against this possibility, however, a CB_1_ antagonist failed to show any antipsychotic effect on patients with schizophrenia [205]. An important correlation to mention about the antipsychotic effects of CBD is the increase in serum of AEA in patients who received CBD and who showed a reduction in psychotic symptoms [130].

### 3.6. CBD Multi-Targets Implicated in Anti-Addictive Action

Systemic evidence has demonstrated that CBD has no potential for abuse since it lacks rewarding effects and did not induce withdrawal-related signs after repeated administration [206,207]. However, the “anti-addictive” actions of CBD have been described in some substance use disorders [172]. In a recent paper, the possible mechanisms underlying the “anti-addictive” potential of CBD involve dopaminergic, opioidergic, endocannabinoid, serotonergic, and glutamatergic systems [172]. However, from a pharmacological approach, the involvement of the 5-HT_1A_R in the effects of CBD on drug-induced reward is the only one that has been demonstrated. The intra-dorsal raphe injection of a 5-HT_1A_R antagonist abolished the CBD-mediated inhibition of the reward-facilitating effect of morphine measured in the intracranial self-stimulation (ICSS) paradigm [171], blocked the effects on ethanol self-administration, in combination with naltrexone, [206] and attenuated CBD-mediated reduction of cocaine self-administration [208].

### 3.7. CBD Multi-Targets Implicated in Alzheimer’s Disease

CBD has neuroprotective properties against the neurotoxic effects of amyloid beta peptide (Aβ) in cell culture and cognitive behavioral models of neurodegeneration that mimic AD symptoms seen in patients [209]. First, CBD seems to activate the PPARγ and at the same time modulate the GPR3 and GPR6 as an inverse agonist. CBD has been shown to favor the activation of PPARγ, reducing inflammation in astrocytes and increasing the survival and neurogenesis of the rat and mouse hippocampus produced by neurotoxicity of Aβ [50,186,189]. The reduction of Aβ expression is also found in in vitro assays on human neuroblastoma SH-SY5Y [187]. Moreover, an overexpression of GPR3 in the brains of AD patients has been reported. While the genetic deletion of Gpr3 gene alleviates the cognitive deficits in AD mice models, CBD acts as inverse agonist on GPR3 [184,185,188]. On the other hand, GPR6 is repressed by Aβ, and its expression is restored by complement protein C1q-induced neuroprotection against Aβ in the hippocampal region in a mouse model [183]. CBD modulates the Kv_2.1_ channel, which in the presence of an inflammatory and oxidative environment produced by the accumulation of Aβ, causes the Kv_2.1_ oxidation, which contributes to cognitive impairment in a mice model of AD [181,182]. In this case, CBD must protect the Kv_2.1_ channel from oxidative damage due to its antioxidant properties. Currently, AChE and BChE enzymes are one of the main targets for AD therapies, these enzymes are inhibited in vitro by CBD [128]. However, the human serum concentration of CBD for the maximum approved dose for Epidiolex (20 mg/kg/day) is estimated to be between 0.3–3.2 μM in the brain [78]. It is likely that at this dose, CBD has no effect on the enzymes AChE and BChE associated with the development of AD.

CBD has a favorable safety profile in humans [18,20], in rats [210], and in dogs [211]. A recent pharmacovigilance study concluded that CBD is a safe compound that causes only rare adverse reactions [212]. However, like any other drug, it is not innocuous; the adverse effects reported in humans so far could vary in severity and include somnolence, fatigue, gastrointestinal ailments, such as diarrhea, vomiting, and changes in appetite/weight, among others [213]. Some of these adverse effects have been addressed experimentally: for example, drowsiness, a common side effect for CBD, in a driving simulation test, indicated that its deleterious effects are minimal, with slightly more collisions and slower brake reaction times [214]. Certain toxicity has been described in preclinical in vitro and in vivo toxicology studies [213]. For instance, CBD-rich cannabis extract causes hepatotoxicity in mice [215]. However, the translation of those results to be applicable to humans has not been bluntly stated, and this seems to be related with the doses range utilized. For example, it has been described that CBD administered between 8–10 µM induces the senescence of human Sertoli cells in vitro, by reducing the transcription of genes involved in cell division, but this concentration is above the average serum concentration of CBD in adults who consumed 1500 mg of daily CBD in a week (0.9 to 2.3 µM)[216]. Finally, CBD action on hepatic enzymes needs to be consider since it can modify the pharmacokinetics of other drugs. A pharmacovigilance study concluded that CBD–drug interactions must be highly considered, since CBD is indicated in people with ongoing treatments, some of them as aggressive as chemotherapy [212].

## 4. Conclusions

In this review, we identified around 56 different neurological molecular targets of CBD including, enzymes, ion channels, ionotropic, and metabotropic receptors. It is crucial to understand CBD’s mechanism of actions in order to ensure the safe use of this secondary metabolite as a therapeutic agent. In this work, we analyzed the multi-target nature of CBD and integrated the information available from in vitro and in vivo studies; we found that there is a little correlation between in vivo, in vitro studies, and clinical trials. The natural product CBD has proven to be a safe agent, because even at the highest doses used to treat neuropsychiatric diseases, it does not reach the serum concentrations necessary to produce long-term toxicity.

Most of the research groups dedicated to cannabinoids are focused on psychoactive cannabinoids, which has made CBD and others phytocannabinoids research an emergent field, and therefore there is room for opportunities to fill in the gap on CBD’s pharmacology. Future studies could be directed on the research of CBD derivatives to enhance specificity on particular targets and in the identification of molecular targets to differentiate, for example, between the effect of CBD on sleep and anxiolytic effect and the anti-emetic and its hypotensive role. Currently there are several clinical trials on the therapeutic effect of CBD in patients with Parkinson’s disease (NCT02818777 and NCT03582137).

## Figures and Tables

**Figure 1 molecules-28-03271-f001:**
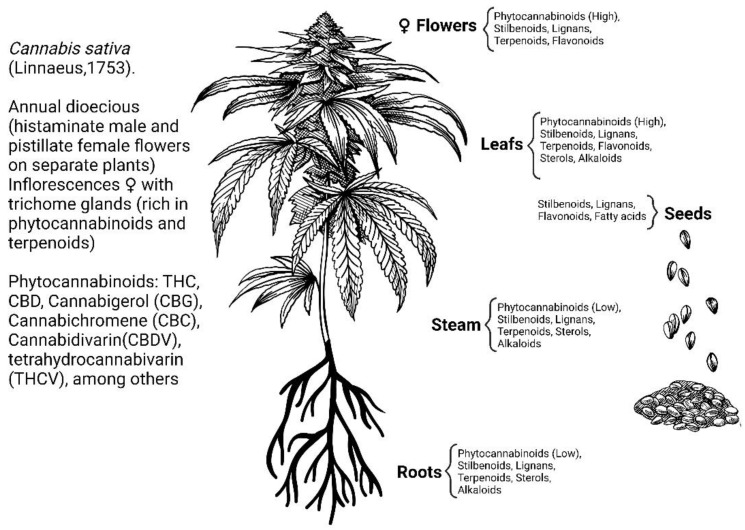
Secondary metabolites of *Cannabis sativa* L. [5].

**Figure 2 molecules-28-03271-f002:**
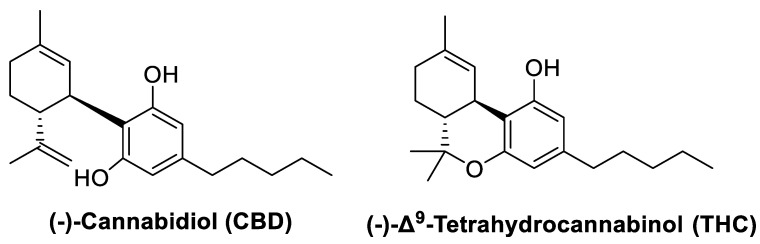
Chemical structures of cannabidiol and tetrahydrocannabinol.

**Figure 3 molecules-28-03271-f003:**
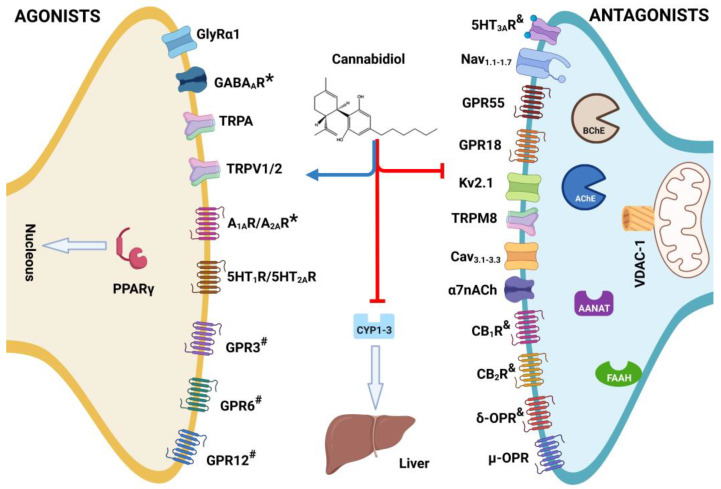
Structure of cannabidiol (CBD) and its role such as agonist or * = positive allosteric modulator (PAM), antagonist, # = inverse agonist, and & = negative allosteric modulator (NAM) on multi-target receptors and enzymes associates with developed of diseases in vivo. Created with BioRender.com.

**Figure 4 molecules-28-03271-f004:**
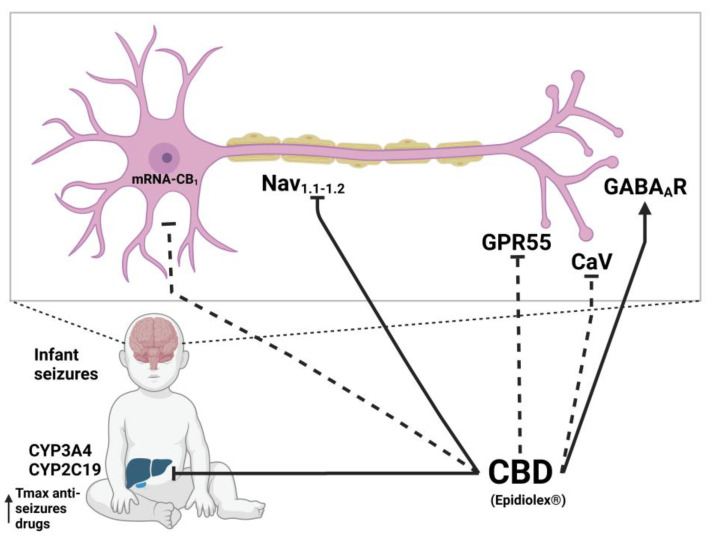
Molecular mechanism of CBD on epilepsy. Dotted lines refer to mechanisms recently suggested as probable based on in vitro experiments, which require verification on in vivo models. Created with Biorender.com.

**Figure 5 molecules-28-03271-f005:**
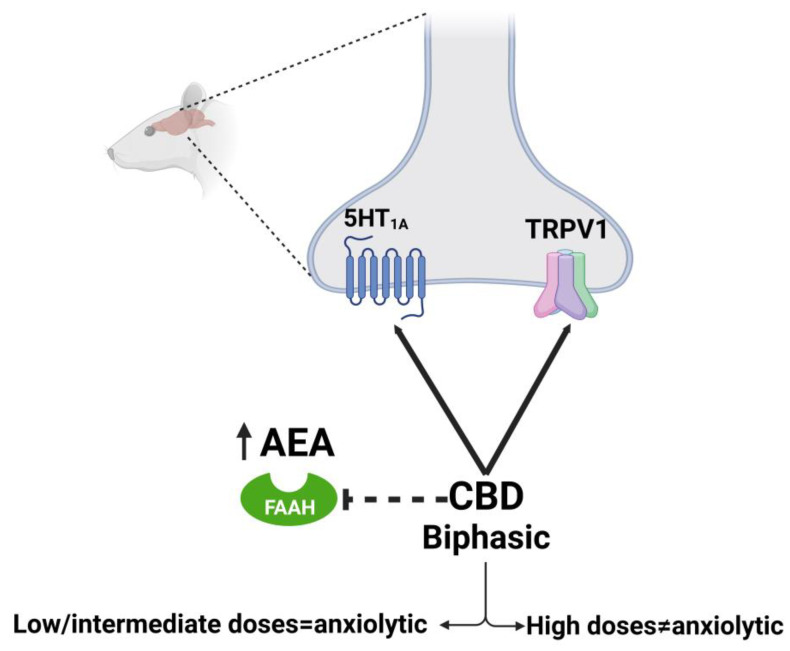
Molecular mechanism of CBD on anxiety described in rat model. Dotted lines refer to hypothesized mechanisms. Created with Biorender.com.

**Table 2 molecules-28-03271-t002:** CBD such as an agonist or PAM of receptors in the nervous system.

Receptor Type	Activity	EC_50_ (µM)	Disease Model	Tissue Expression	Cited
GlyRα1	Agonist	12.3	Persistent inflammatory and neuropathic pain in rodents.	Brainstem and spinal cord.	[39,40]
GlyRα2	-	-	Neuropathic pain.	Dorsal horn.	
GlyRα3	PAM	-	Antinociceptive process and inflammatory pain.	Expressed in the superficial layer of the spinal dorsal horn.	
GABA_A_R	PAM	2.4	Model of kainic acid-induced temporal lobe epilepsy in rats.	Expressed on synaptic and extrasynaptic sites of CNS.	[41,42,43]
TRPV1	Agonist	0.1–1.0	Seizures in TRPV1-KO mice. Phagocytic capacity of microglia for clearance of β-amyloid in AD.	Dorsal root ganglia neurons, trigeminal ganglia, peripheral afferent fibers, and specific on nociceptive sensory endings.	[43,44,45]
TRPV2	Agonist	1.25–3.7	Phagocytic capacity of microglia for clearance of β-amyloid in AD.	Dorsal root ganglia neurons, trigeminal ganglia, peripheral afferent fibers, and especially on nociceptive sensory endings.	[43,44,45]
TRPV3	Agonist	3.7	-	-	[44]
TRPV4	Agonist	0.8	-	Brain, cutaneous A and C fibers and tongue.	[44,46]
TRPA1	Agonist	0.11		Co-expressed with TRPV1 in nociceptive neurons of the PNS.	[47]
A_1A_	PAM	-	Neuropathic pain and inflammation models shown hyperalgesia or allodynia.	Widespread in the brain.	[48,49]
A_2A_	PAM	-		Highly expressed in the striatum.	
PPARγ	Agonist		Rat model of AD; mice model of MS.	Adipose tissue and macrophages.	[50,51,52]

AD = Alzheimer’s disease; CNS = Central nervous system; MS = Multiple sclerosis; PAM = Positive allosteric modulator; PNS = Peripheric nervous system.

**Table 5 molecules-28-03271-t005:** Targets of CBD on neurological diseases.

Disease	Doses (mg/Kg/Day)	Agonist	Antagonist	Cited
Anxiolytic	1–50	TRPV1-2	CB_1_	[62,110,111,112,152,153,154,155,156,157,158,159,160]
Antidepressant		5-HT_1-2A_R	FAAH	
Antinociceptive	3–30	GABA_A_R	CB_1_ TRPM8 Ca_V3.1–3.3_ δ-OPR µ-OPR GPR55 GPR18 FAAH	[40,91,100,126,161,162,163,164,165,166,167,168]
		TRPV1		
		TRPA1		
		A_1A_R		
		GlyRα1		
		PPARγ		
		5HT_1A_R		
Epilepsy	5–20	GABA_A_R	Nav_1.1–1.7_	[41,71,72,73,74,75,163,169,170]
			CaV_3.1–3.3_	
			CB_1_	
			GPR55	
			CYP3A4	
			CYP2C19	
Antiaddictive effects	400–800	GPR3	δ-OPR	[171,172]
		5-HT1A		
Schizophrenia	600–1000	GPR6	CB_1_	[100,101,173,174]
			δ-OPR	
			µ-OPR	
			FAAH	
Anti-inflammatory, immunosuppressive		TRPV1	α7nAChR VDAC-1 CB_2_ δ-OPR µ-OPR GPR18 FAAH	[45,76,87,122,129,174,175,176,177,178,179,180]
		TRPV2		
		TRPA		
		A_1A_R		
		A_2A_R		
		PPARγ		
Reversing cognitive of Alzheimer’s disease	50	GPR3	Kv_2.1_	[50,128,181,182,183,184,185,186,187,188,189]
		GPR6	AChE	
		PPARγ	BChE	

## Data Availability

Data sharing not applicable.

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
