# Peer review of "The Polypharmacological Effects of Cannabidiol"

_molecules, 2023, doi:10.3390/molecules28073271_

Round 1

Reviewer 1 Report

I have read the article entitled “Deciphering the polypharmacological effects of cannabidiol”. At this moment, I have a lot of doubts connected with this title. Based on many publications (209) preparing this article, the authors collected the information. Other people did the process of deciphering.

I do not support this type of publication because there is no intellectual and experimental contribution of the authors. I appreciate the effort devoted to the preparation of this work. But, the collection of information based on  209 references makes this work difficult to read. For the reviewer, viewing at least some of these works within 7 days is impossible.

In addition, out of 209 references, only one (115) is listed as a co-author (J. Castillo – Arellano), one of the authors of this work. Please explain the author's experience in this subject since they focused on preparing such work.

In conclusion, the authors wrote: “However, there are still  many gaps in the information, which do not allow a complete understanding of the molecular  mechanism by which therapeutic effects occur, since there is little correlation between in vivo, in vitro studies and clinical trials of CBD.”

Maybe the authors should focus on practical explanations for these problems.

I cannot recommend this work for publication. 

Author Response

Comment #1. I have read the article entitled “Deciphering the polypharmacological effects of cannabidiol”. At this moment, I have a lot of doubts connected with this title. Based on many publications (209) preparing this article, the authors collected the information. Other people did the process of deciphering.

Answer: Thank you for the comment, we understand that the “deciphering” does not like to the reviewer, and we have removed the word “Deciphering” from the title to avoid controversy.

Comment #2. I do not support this type of publication because there is no intellectual and experimental contribution of the authors. I appreciate the effort devoted to the preparation of this work. But, the collection of information based on 209 references makes this work difficult to read. For the reviewer, viewing at least some of these works within 7 days is impossible.

Answer: The main point of this review is to give the scientific community a glance of CBD’s studies. To be honest, I do not concur with the reviewer’s comment. I think that there is great validity in reviews in general (no matter the number of references).

Comment #3. In addition, out of 209 references, only one (115) is listed as a co-author (J. Castillo – Arellano), one of the authors of this work. Please explain the author's experience in this subject since they focused on preparing such work.

Answer: Related to the lack of authorship’s experience on the topic and the absence of their references in the review. We believed that it is not pertinent to auto cite much saying that, following the reviewer comments and on the same page on the experience of Dr. Castillo-Arevalo which is greatly appreciated (one reference on cannabinoids, pointed by the reviewer), the experience of Dr Canseco-Alba can be summarized in several publication on the topic, she has been working at least last 10 years on the pharmacology of AEA, 2-AG and phytocannabinoids on endocannabinoid system using different animal models. She also has been working with Dr. Onaivi, -a prominent figure on the endocannabinoid system, on CB1 and CB2 knock-out rodent models. She is knowledgeable on the topic some of her publications are found in PMID:35211038; PMID:33371336; PMID:30508607; PMID:30439451; PMID:31474840; PMID:29991708; PMID:31480324; PMID:29234141; PMID:27856203; PMID:26226145; PMID:24671517; PMID:22906359. In the other hand Dr. Cutler is a renowned researcher/Professor with high federal NIH-funding on the area of cannabinoid/opioids. Drs. Cutler and Leon are experts in the isolation, structural elucidation of natural products including phytocannabinoids, and in vitro and in vivo pharmacology, some of their works on cannabinoids have been published, others not. A few of their combined publications in this specific topic is summarized in: https://doi.org/10.1016/j.arabjc.2021.103545; PMID: 26035635; PMID:25419092; PMID:24288291; PMID:21667972.

We have increased the number of auto-citations as was suggested by the reviewer.

Comment #4. In conclusion, the authors wrote: “However, there are still many gaps in the information, which do not allow a complete understanding of the molecular mechanism by which therapeutic effects occur, since there is little correlation between in vivo, in vitro studies and clinical trials of CBD.” Maybe the authors should focus on practical explanations for these problems.

Answer: Holistically is way difficult to understand everything about a specific metabolite, and there are in fact some gaps or/and ongoing research. We included in the discussion and conclusion sections paragraphs about the gaps in CBD pharmacology and about specific problems that require practical explanation. Also, we added some figures intended to illustrate the mechanistic explanations for CBD effects.

Reviewer 2 Report

1.       Author may also include FDA approved cannabis-based product such as synthetic cannabis-related drug products and cannabis-derived compound.

2.       Toxicity study of CBD need to be discussed.

3.       Grammatical errors in the manuscript which need revision.

4.       Include references from year 2023 and discuss (although last submission date was 31 October, 2022)

5.       Similarity matching needs to be corrected e.g Line 397-404; 463-465; 468-470;482-483; 545-548 .

6.       Conclusion should draw the theme of special issue Improvements and Opportunities on Natural Products for Novel Drug Discovery

7.       Concluding point is missing in Abstract.

Author Response

Comment #1. Author may also include FDA approved cannabis-based product such as synthetic cannabis-related drug products and cannabis-derived compound.

Answer: Thank to the reviewer for the comments, we have included a table with relevant information about the FDA-approved cannabis drugs (Table 1).

Comment #2. Toxicity study of CBD need to be discussed.

Answer: We included a toxicology paragraph in the discussion section, and include some references about it

Comment #3. Grammatical errors in the manuscript which need revision.

Answer: We reviewed the manuscript and made modifications.

Comment #4. Include references from year 2023 and discuss (although last submission date was 31 October 2022)

Answer: We found 229 papers of 2023 about CBD, but only we included 5 relevant recent papers (2023) and those were included in toxicology and inflammatory sections.

Comment #5. Similar matching needs to be corrected e.g Line 397-404; 463-465; 468-470;482-483; 545-548.

Answer: Thank you so much to the reviewer, the similarity in those paragraphs was corrected

Comment #5. Conclusion should draw the theme of special issue Improvements and Opportunities on Natural Products for Novel Drug Discovery.

Answer: The opportunity for CBD in Drug Discovery is denoted in along the discussion and conclusions and are directly aligned to the focus of the special issue “Improvements and Opportunities on Natural Products for Novel Drug Discovery”.

Comment #6. Concluding point is missing in Abstract.

Answer: We included the conclusion in the abstract (last phrase)

Reviewer 3 Report

I read the paper carefully and realized that it needs a major revision. About this manuscript, there are some ambiguous points which should be addressed and clarified in the next revision as being listed below:

The plant authority should be given when revising this manuscript again all through the revision since it is of prime importance in such sorts of studies in the literature.

The keywords should be sorted alphabetically.

If possible, I suggest inserting a list of the main abbreviations used. In this sense, the relevant abbreviations should in line in alphabetical order and be separated using a semicolon from each other. This will make the manuscript easier to read.

Remove extra spaces before and after the sign “-“.

The authors are expected to give a mechanistic approach when discussing different topics and aspects related to their study in due parts of their revised article.

The authors are expected to add a new phrase concerning the presence of a large and wide spectrum of natural compounds constituting groups in their different organs referring to the following articles:

DOI: 10.30495/tpr.2022.1951155.1243

DOI: 10.30495/tpr.2022.1954612.1248

In this connection, no need to mention the name of the plant genus and just the authors should elaborately imply the aforementioned topics concisely to make a more valuable approach relating to the study.

The authors have implied some promising and therapeutic pharmacological activities of of cannabidiol but without giving a mechanistic approach to justify these behaviors and this is so important to give a deeper insight into the topic of research, of course. In other words, a better mechanistic approach is anticipated when justifying the relevant observed pharmacological activities and this is necessary in the majority of cases providing reliable and scientific proofs.

Author Response

Comment #1. I read the paper carefully and realized that it needs a major revision. About this manuscript, there are some ambiguous points which should be addressed and clarified in the next revision as being listed below: The plant authority should be given when revising this manuscript again all through the revision since it is of prime importance in such sorts of studies in the literature.

Answer: We added the plant authority of Cannabis sativa (Linneo, 1753) for first mention and C. sativa L for the rest of the document.

Comment #2. The keywords should be sorted alphabetically.

Answer: We have been modifying the keywords.

Comments #3. If possible, I suggest inserting a list of the main abbreviations used. In this sense, the relevant abbreviations should in line in alphabetical order and be separated using a semicolon from each other. This will make the manuscript easier to read.

Answer: We are completely in agreement with the reviewer, a section on the abbreviations is included in the revised version

Comments #4. Remove extra spaces before and after the sign “-“.

Answer: Thank you, we removed the extra spaces.

Comment #5. The authors are expected to give a mechanistic approach when discussing different topics and aspects related to their study in due parts of their revised article.

Answer: We included two figures of the most important activities of CBD. One figure about the molecular mechanism of CBD on epilepsy and one figure about of molecular mechanism of anxiolytic properties of CBD.

Comment #6. The authors are expected to add a new phrase concerning the presence of a large and wide spectrum of natural compounds constituting groups in their different organs referring to the following articles:

DOI: 10.30495/tpr.2022.1951155.1243

DOI: 10.30495/tpr.2022.1954612.1248

In this connection, no need to mention the name of the plant genus and just the authors should elaborately imply the aforementioned topics concisely to make a more valuable approach relating to the study.

Answer: Thank you to the reviewer, we included one figure about natural products of Cannabis sativa L (Figure 1) as summary for the natural products found in different part of the plant. The discussion was kept on the topic, and we avoid deviating from the main point.

Comment #7. The authors have implied some promising and therapeutic pharmacological activities of cannabidiol but without giving a mechanistic approach to justify these behaviors and this is so important to give a deeper insight into the topic of research, of course. In other words, a better mechanistic approach is anticipated when justifying the relevant observed pharmacological activities and this is necessary in the majority of cases providing reliable and scientific proofs.

Answer:  We completely concurred with the reviewer comments, and we included some discussion about the mechanistic data available and we also added a couple figures of the most important activities of CBD as we mentioned before.

Round 2

Reviewer 1 Report

I am not convinced about this type of article, but I appreciate the effort devoted to preparing this article.

After the revision, I can recommend this manuscript for publication.  

Reviewer 2 Report

All the comments are included in the review.
